# Discrete-Time First-Order Plus Dead-Time Model-Reference Trade-off PID Control Design

**Ryo Kurokawa** [1], **Takao Sato** [1,*] , **Ramon Vilanova** [2] and **Yasuo Konishi** [1]

[1] Department of Mechanical Engineering, Graduate School of Engineering, University of Hyogo, Hyogo 671-2280, Japan

[2] Department of Telecommunications and Systems Engineering, Universitat Autònoma de Barcelona, 08193 Barcelona, Spain

* Correspondence: tsato@eng.u-hyogo.ac.jp; Tel.: +81-792674983

**Abstract:** The present study proposes a novel proportional-integral-derivative (PID) control design method in discrete time. In the proposed method, a PID controller is designed for first-order plus dead-time (FOPDT) systems so that the prescribed robust stability is accomplished. Furthermore, based on the control performance, the relationship between the servo performance and the regulator performance is a trade-off relationship, and hence, these items are not simultaneously optimized. Therefore, the proposed method provides an optimal design method of the PID parameters for optimizing the reference tracking and disturbance rejection performances, respectively. Even though such a trade-off design method is being actively researched for continuous time, few studies have examined such a method for discrete time. In conventional discrete time methods, the robust stability is not directly prescribed or available systems are restricted to systems for which the dead-time in the continuous time model is an integer multiple of the sampling interval. On the other hand, in the proposed method, even when a discrete time zero is included in the controlled plant, the optimal PID parameters are obtained. In the present study, as well as the other plant parameters, a zero in the FOPDT system is newly normalized, and then, a universal design method is obtained for the FOPDT system with the zero. Finally, the effectiveness of the proposed method is demonstrated through numerical examples.

**Keywords:** PID control; model-based design; discrete time system; sensitivity function; robust stability; FOPDT system

## 1. Introduction

Proportional-integral-derivative (PID) [1–9] control has few tuning parameters: proportional gain, integral time, and derivative time, and its structure is simple. Hence, PID control has been widely used in industry, and numerous tuning methods have been proposed.

In the model-based approach, optimal tracking and robust stability are achieved. However, the control system must be redesigned whenever the controlled system is changed. As a simple model-based optimal design method, Ziegler and Nichols proposed the step response method (ZN method) [10]. Using the ZN method, the PID parameters are decided based on the step response trajectory so that the tracking performance is optimized. However, the robust stability is not taken into account [11,12]. Therefore, robust control designs have been proposed [13,14]. Using robust design methods, robust stability is obtained, but the tracking performance is not optimized. In [15], $H_2$ optimal design was proposed using internal model control (IMC) design, and hence, stability was assured. Although robust PID control systems are also designed in the discrete time domain [16,17], the stability margin cannot be prescribed.

The relationship between the tracking performance and the robust stability is a trade-off relationship [18]. Therefore, trade-off PID design methods have been proposed [19,20]. In the trade-off design methods, the PID parameters are decided such that the tracking performance is optimized subject to the prescribed robust stability, where servo or regulator mode is selected for performance optimization. A trade-off optimization approach has been designed in the continuous time domain that consists of a first-order plus dead-time (FOPDT) system [21,22], a second-order plus dead-time (SOPDT) system [23,24], and a two degrees-of-freedom (2DOF) system [25,26].

For a discrete time system, trade-off approaches have been proposed [27,28]. Discrete-time methods are useful for controlling discrete time systems, in which the controller is implemented with a digital computer. However, the robust stability was not assigned in [27], and the controlled plant was restricted to a non-zero system [28]. Therefore, in the present study, a new discrete time domain approach is proposed, in which the constraint is relaxed and the prescribed robust stability is accomplished. Specifically, the FOPDT model with a zero is normalized, and a universal design method is obtained. As a result, the trade-off design strategy is available for a large class of discrete time systems. Finally, the effectiveness of the proposed method is demonstrated through numerical examples.

## 2. Control System

The continuous time FOPDT transfer function is described as follows:

$$P(s) = \frac{K}{Ts+1}e^{-Ls} \tag{1}$$

where $K$ is the gain, $T$ is the time constant, and $L$ is the dead time. In the present study, the discrete time control model is designed with sampling interval $T_s$, and hence, the continuous time model is expressed as follows:

$$P_d(z^{-1}) = \frac{b_0 + b_1 z^{-1}}{1 - a_1 z^{-1}} z^{-(d+1)} \tag{2}$$

where $a_1$, $b_0$, and $b_1$ are the coefficient parameters, $d$ is the dead-time in the discrete time, and $z^{-1}$ denotes the backward shift operator. The discrete time plant parameters correspond to the continuous time plant parameters as follows:

$$a_1 = e^{-\frac{T_s}{T}} \tag{3}$$

$$b_0 = K\left(1 - a_1 e^{\frac{L_0}{T}}\right) \tag{4}$$

$$b_1 = K\left(a_1 e^{\frac{L_0}{T}} - a_1\right) \tag{5}$$

$$d = L_1 \tag{6}$$

where a non-negative integer $L_1$ satisfies the following equation:

$$L = L_1 T_s + L_0 \ \ (L_0 < T_s)$$

Therefore, the input/output relationship in discrete time is given as follows:

$$y(k) = P_d(z^{-1})u(k) \tag{7}$$

where $y(k)$ is the system output (plant output) and $u(k)$ is the control input.

The present study proposes a new trade-off design method for the following PID control law:

$$
\begin{aligned}
u(k) &= C_e(z^{-1})e(k) - C_y(z^{-1})y(k) \\
C_e(z^{-1}) &= K_p\left\{1 + \frac{T_s}{T_i(1-z^{-1})}\right\} \\
C_y(z^{-1}) &= K_p\left\{\frac{T_d(1-z^{-1})}{T_s}\right\} \\
e(k) &= r(k) - y(k)
\end{aligned}
\tag{8}
$$

where $r(k)$ is the reference input and $K_P$, $T_i$, and $T_d$ are the proportional gain, the integral time, and the derivative time, respectively. A block diagram of the control system is shown in Figure 1, in which the control input is disturbed by disturbance $d(k)$. Equation (8) is the discrete version of the next continuous time control law:

$$
\begin{aligned}
U(s) &= K_p\left\{\left(1 + \frac{1}{T_i s}\right)E(s) - T_d s Y(s)\right\} \\
E(s) &= R(s) - Y(s)
\end{aligned}
\tag{9}
$$

where $\mathcal{L}[\cdot]$ means the Laplace transform, $U(s) = \mathcal{L}[u(t)]$, $Y(s) = \mathcal{L}[y(t)]$, and $R(s) = \mathcal{L}[r(t)]$.

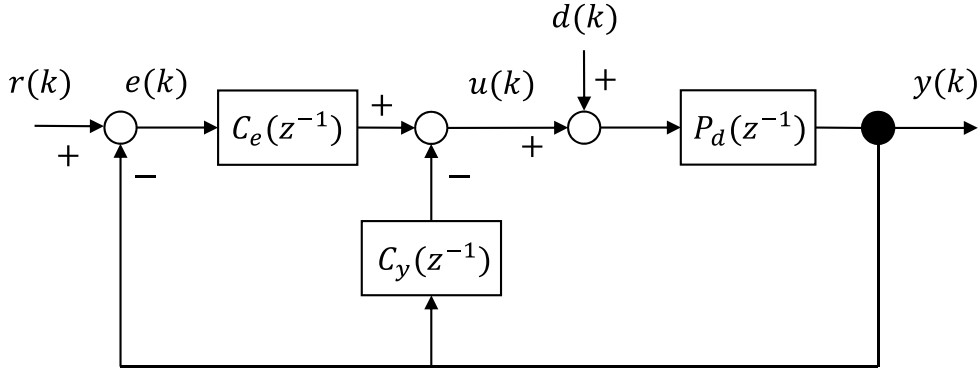

**Figure 1.** Discrete-time PID control system.

The closed-loop systems from $r(k)$ and $d(k)$ to $u(k)$ and $y(k)$, respectively, are obtained as follows:

$$
u(k) = \frac{C_e(z^{-1})}{1 + C_d(z^{-1})P_d(z^{-1})}r(k) - \frac{C_d(z^{-1})P_d(z^{-1})}{1 + C_d(z^{-1})P_d(z^{-1})}d(k)
\tag{10}
$$

$$
\begin{aligned}
y(k) &= \frac{C_e(z^{-1})P_d(z^{-1})}{1 + C_d(z^{-1})P_d(z^{-1})}r(k) + \frac{P_d(z^{-1})}{1 + C_d(z^{-1})P_d(z^{-1})}d(k) \\
C_d(z^{-1}) &= C_e(z^{-1}) + C_y(z^{-1})
\end{aligned}
\tag{11}
$$

From the above equations, the relationship between the reference tracking and the disturbance rejection is a trade-off relationship.

## 3. Design Objective

In the proposed method, the PID parameters of the discrete time control law are designed based on a constrained optimization problem. The constrained optimization problem consists of the constraint condition as the prescribed stability margin and the objective function defined by the index with respect to the servo and regulator performances, respectively. The constraint condition and the objective function are defined in Sections 3.1 and 3.2, respectively.

### 3.1. Constraint Condition

The constraint condition is used to obtain the stability margin. In the proposed method, the stability margin is designed using the following sensitivity function:

$$S_f(z^{-1}) = \frac{1}{1 + C_d(z^{-1})P_d(z^{-1})} \tag{12}$$

Using the sensitivity function, the constraint condition is defined as follows:

$$|M_s - M_s^d| = 0 \tag{13}$$

$$M_s = \max_{\omega} |S_f(e^{-j\omega})| \tag{14}$$

where $M_s$ denotes the maximum value of the sensitivity function $S_f(e^{-j\omega})$ and $M_s^d$ is the designed value of $M_s$.

The design range of $M_s^d$ is recommended to be from 1.4–2.0 [4]. The relationship between $M_s^d$ and the stability margin is in inverse proportion.

### 3.2. Objective Function

The objective function is defined based on the sum of absolute errors (SAE) as follows:

$$J_d = \sum_{k=1}^{\infty} |r(k) - y(k)| \tag{15}$$

In a one-degree-of-freedom (1DOF) system, the relationship between the servo and regulation performances is a trade-off relationship. Therefore, the objective functions in the reference tracking optimization and the disturbance rejection optimization are given by Equations (16) and (17), respectively:

$$J_d^s = \sum_{k=1}^{\infty} \left| r^s(k) - \frac{C_e(z^{-1})P_d(z^{-1})}{1 + C_d(z^{-1})P_d(z^{-1})} r(k) \right| \tag{16}$$

$$J_d^r = \sum_{k=1}^{\infty} \left| r^d(k) - \frac{P_d(z^{-1})}{1 + C_d(z^{-1})P_d(z^{-1})} d(k) \right| \tag{17}$$

where $r^s(k) = 1$, $r(k) = 1$, $r^d(k) = 0$, and $d(k) = 1$.

In the discussed control system, the stability and the tracking performance are simultaneously decided because they are in a trade-off relationship.

## 4. Proportional-Integral-Derivative Parameter Design

In order to obtain a universal design method for arbitrary FOPDT models, first, in Section 4.1, the plant model and the PID control law are normalized. Second, in Section 4.2, optimal PID parameters are obtained subject to the pre-established stability margin. Third, in Section 4.3, the parameter tuning rule is derived based on the optimal parameters calculated in Section 4.2. Finally, the algorithm for the proposed design method is summarized in Section 4.4.

### 4.1. Normalization

The plant model and the control law are normalized to obtain a universal design method. The normalization parameters for the plant model, $\tau_0$ and $\tau_a$, are given as follows:

$$\tau_0 = -d \log a_1 + \log\left(\frac{b_0 a_1 + b_1}{a_1(b_0 + b_1)}\right) \tag{18}$$

$$\tau_a = -\log a_1 \tag{19}$$

Moreover, the normalization parameters for the control law, $\kappa_p$, $\tau_i$, and $\tau_d$, are given as follows:

$$\kappa_p = \frac{b_0 + b_1}{1 - a_1} Kp \tag{20}$$

$$\tau_i = -\frac{T_i \log a_1}{T_s} \tag{21}$$

$$\tau_d = -\frac{T_d \log a_1}{T_s} \tag{22}$$

The derivation of the normalization parameters is shown in Appendix A.

The expression of the plant model using the normalization parameters has merits for optimization analysis and controller design. Since the optimization problem is stated in terms of just two parameters $\tau_0$ and $\tau_a$, the analysis and elaboration of coefficient parameters are simplified. Furthermore, the controller parameters are tuned independent of the process gain and time constant.

### 4.2. Optimization

The controller parameters are optimized for the servo and regulation operations, respectively, subject to the established stability margin.

As the optimization tool, the fmincon function in MathWorks MATLAB software was used, where the prescribed robust stability is set: $M_s^d \in \{1.4, 1.6, 1.8, 2.0\}$, and the optimal controller parameters are calculated based on the normalization parameters for the controlled plant: $\tau_0 \in \{0.3, \cdots, 1.7\}$ and $\tau_a \in \{0.01, \cdots, 0.1\}$. As an example, the optimized controller parameters for $M_s^d = 1.4$ in the servo are shown in Figure 2.

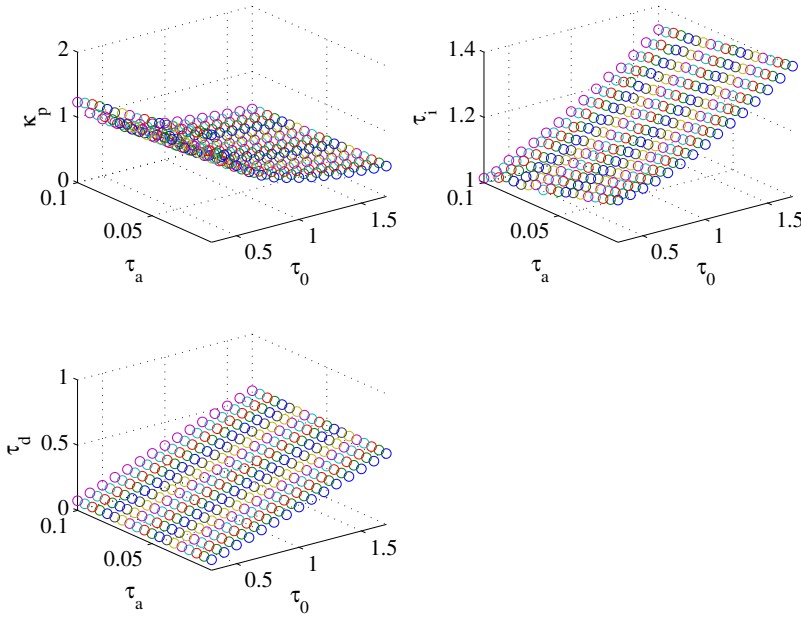

**Figure 2.** Relationships among optimized controller parameters and the normalized parameters $\tau_0$ and $\tau_a$ in servo design.

### 4.3. Controller Parameter Decision

The tuning rule for the controller parameters is proposed in Section 4.3.1, and the prescribed robust stability using the proposed parameter tuning rule is evaluated in Section 4.3.2.

### 4.3.1. Tuning Rule

On the pair of plant parameters $\tau_0$ and $\tau_a$ with the established robust stability $M_s^d$, the controller parameters are expressed in terms of the normalization parameters for the plant model as:

$$\kappa_p = \alpha_0 + \alpha_1 \tau_0^{\alpha_2} \tag{23}$$

$$\tau_i = \beta_0 + \beta_1 \tau_0 + \beta_2 \tau_0^2 + \beta_3 \tau_0^3 \tag{24}$$

$$\tau_d = \gamma_0 + \gamma_1 \tau_0 + \gamma_2 \tau_0^2 \tag{25}$$

where $\alpha_i (i = 0, 1, 2)$, $\beta_j (j = 0, 1, 2, 3)$, and $\gamma_k (k = 0, 1, 2)$ are defined as follows:

$$\alpha_0 = a_{00} + a_{01} \tau_a \tag{26}$$

$$\alpha_1 = a_{10} + a_{11} \tau_a \tag{27}$$

$$\alpha_2 = a_{20} + a_{21} \tau_a \tag{28}$$

$$\beta_0 = b_{00} + b_{01} \tau_a \tag{29}$$

$$\beta_1 = b_{10} + b_{11} \tau_a \tag{30}$$

$$\beta_2 = b_{20} + b_{21} \tau_a \tag{31}$$

$$\beta_3 = b_{30} + b_{31} \tau_a \tag{32}$$

$$\gamma_0 = c_{00} + c_{01} \tau_a \tag{33}$$

$$\gamma_1 = c_{10} + c_{11} \tau_a \tag{34}$$

$$\gamma_2 = c_{20} + c_{21} \tau_a \tag{35}$$

where the coefficient parameters in Equation (26)–Equation (35) in the servo optimization are shown in Table 1, and those in the regulation optimization are also shown in Table 2.

**Table 1.** Coefficient parameters in Equations (26)–(35) in servo design.

| $M_s^d$ | 1.4 | 1.6 | 1.8 | 2.0 |
|---|---|---|---|---|
| $a_{00}$ | 0.2130 | 0.2778 | 0.3281 | 0.3098 |
| $a_{01}$ | −0.4643 | −0.6376 | −0.8185 | −0.7722 |
| $a_{10}$ | 0.4361 | 0.5803 | 0.6932 | 0.8100 |
| $a_{11}$ | −0.3767 | −0.4236 | −0.3308 | −0.4577 |
| $a_{20}$ | −1.0067 | −1.0169 | −1.0150 | −0.9861 |
| $a_{21}$ | 1.7509 | 1.7951 | 1.9003 | 1.8503 |
| $b_{00}$ | 1.1368 | 1.1451 | 1.2097 | 1.3995 |
| $b_{01}$ | −1.6140 | −1.1310 | −0.7911 | −1.9403 |
| $b_{10}$ | −0.0394 | 0.3152 | 0.4516 | 0.1364 |
| $b_{11}$ | 1.4393 | 0.0802 | −1.2593 | 2.0622 |
| $b_{20}$ | 0.1724 | −0.0447 | −0.1094 | 0.1498 |
| $b_{21}$ | −0.9219 | 0.3521 | 1.6861 | −1.2358 |
| $b_{30}$ | −0.0326 | 0.0265 | 0.0354 | −0.0201 |
| $b_{31}$ | 0.2070 | −0.1725 | −0.5677 | 0.2429 |
| $c_{00}$ | −0.0190 | 0.000066 | 0.0047 | 0.0091 |
| $c_{01}$ | −0.1314 | −0.0898 | −0.0615 | −0.0129 |
| $c_{10}$ | 0.3193 | 0.2819 | 0.3377 | 0.3596 |
| $c_{11}$ | 0.3330 | 0.0381 | 0.0363 | 0.0514 |
| $c_{20}$ | 0.0056 | −0.0100 | −0.0242 | −0.0090 |
| $c_{21}$ | −0.0527 | −0.0124 | 0.0078 | −0.0046 |

**Table 2.** Coefficient parameters in Equations (26)–(35) in regulator design.

| $M_s^d$ | 1.4 | 1.6 | 1.8 | 2.0 |
|---|---|---|---|---|
| $a_{00}$ | 0.2085 | 0.2718 | 0.2999 | 0.3672 |
| $a_{01}$ | −0.6075 | −0.8871 | −0.6490 | −1.4148 |
| $a_{10}$ | 0.4445 | 0.5897 | 0.7267 | 0.7914 |
| $a_{11}$ | −0.3597 | −0.3261 | −0.7568 | −0.1116 |
| $a_{20}$ | −1.0048 | −1.0010 | −0.9840 | −1.0107 |
| $a_{21}$ | 2.4219 | 2.5022 | 2.1738 | 2.7688 |
| $b_{00}$ | 0.2175 | 0.1208 | 0.1676 | 0.1793 |
| $b_{01}$ | 1.0142 | 1.4350 | 0.5152 | 0.5668 |
| $b_{10}$ | 1.3058 | 1.5359 | 1.4478 | 1.3845 |
| $b_{11}$ | −4.3025 | −4.9006 | −1.6551 | −1.4977 |
| $b_{20}$ | −0.7838 | −0.8310 | −0.6531 | −0.4397 |
| $b_{21}$ | 3.7862 | 4.0734 | 0.9992 | 0.8169 |
| $b_{30}$ | 0.2250 | 0.2067 | 0.1519 | 0.0589 |
| $b_{31}$ | −1.0977 | −1.1117 | −0.2245 | −0.1967 |
| $c_{00}$ | −0.0031 | 0.0139 | 0.0152 | 0.0314 |
| $c_{01}$ | 0.0802 | 0.1103 | 0.0765 | 0.1761 |
| $c_{10}$ | 0.4456 | 0.3783 | 0.3607 | 0.3006 |
| $c_{11}$ | 0.3391 | 0.0800 | −0.0139 | −0.3791 |
| $c_{20}$ | −0.0467 | −0.0296 | −0.0374 | −0.0100 |
| $c_{21}$ | −0.1076 | −0.0107 | 0.0186 | 0.2333 |

The controller parameters are optimized in the limited range. Although the range can be expanded, the tuning rule is redesigned when the range is changed.

### 4.3.2. Evaluation of the Prescribed Robust Stability

Non-obtained controller parameters are interpolated, where using the tuning rule proposed in Section 4.3.1, the controller parameters are calculated on the pair of the normalization parameters for the controlled plant: $\tau_0 \in \{0.3, 0.31, \cdots, 1.7\}$ and $\tau_a \in \{0.01, 0.011, \cdots, 1.0\}$, where $M_s^d$ is varied in $\{1.4, \cdots, 2.0\}$. The calculated controller parameters for $M_s^d = 1.4$, merged into the preliminary calculated parameters shown in Figure 2, are shown in Figure 3. The calculated parameters using the proposed tuning rule are sufficiently close to the optimized parameters.

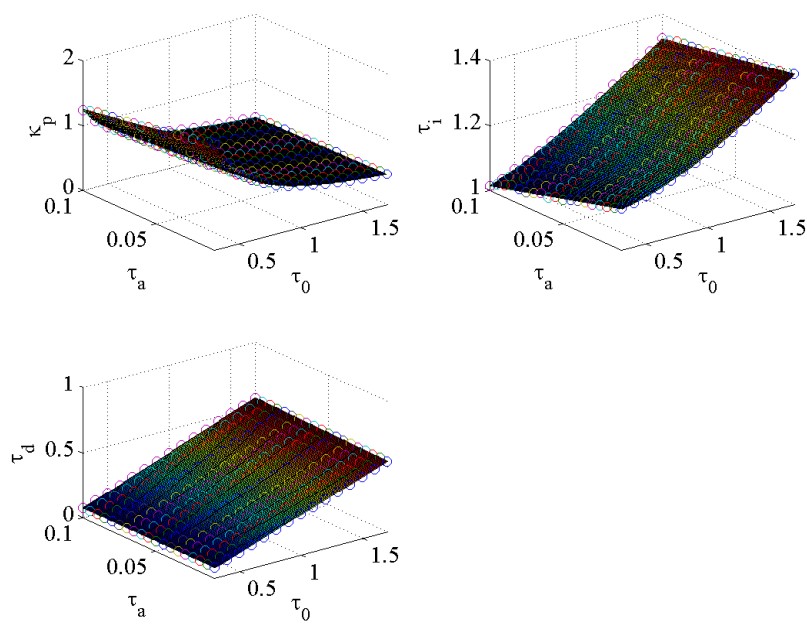

**Figure 3.** Relationship between approximated controller parameters in servo design.

Next, $M_s$ values calculated using the calculated controller parameters in the servo and regulation optimization are shown in Figures 4 and 5, respectively. The minimum and maximum calculated $M_s$ values are summarized in Table 3. Since the errors between $M_s^d$ and $M_s$ are within ±5%, the proposed tuning rule has sufficient precision.

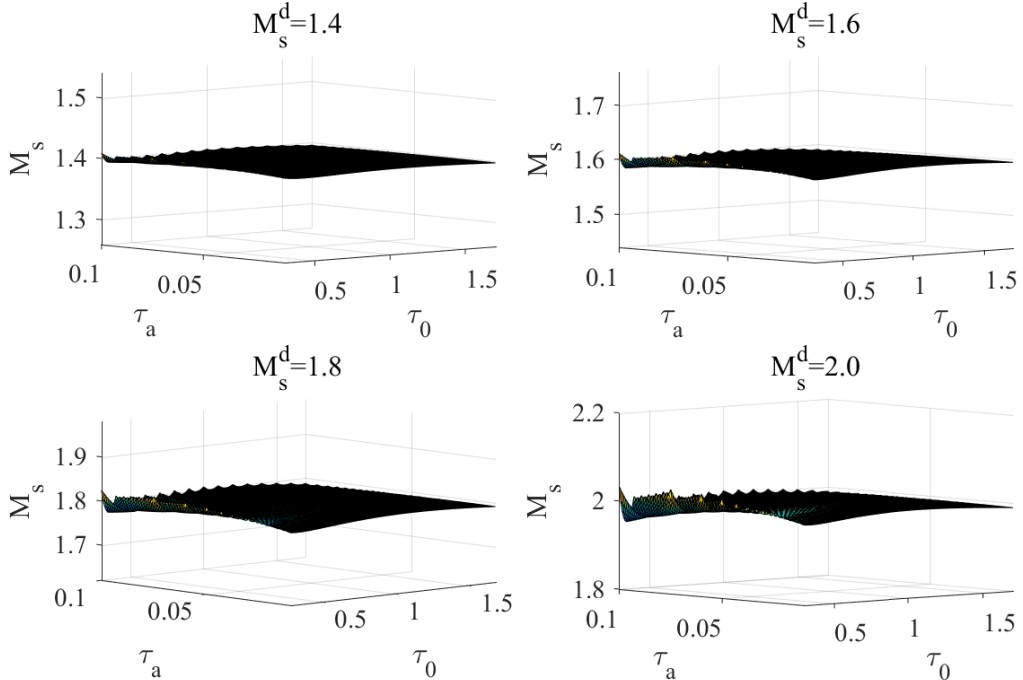

**Figure 4.** Relationships among the obtained $M_s$, $\tau_0$, and $\tau_a$ in servo design.

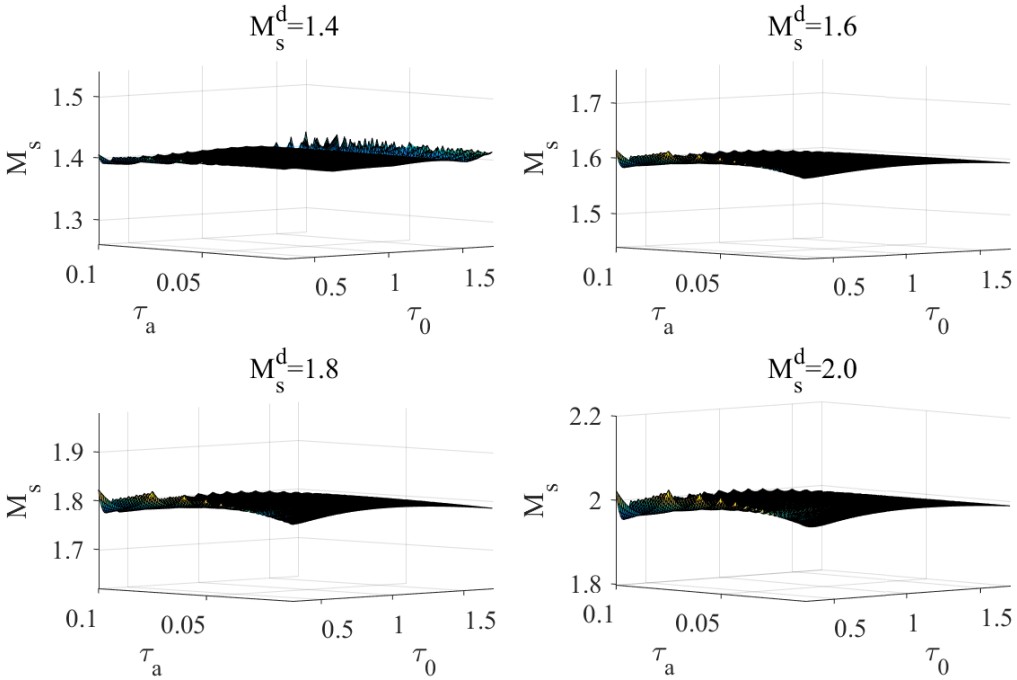

**Figure 5.** Relationships among the obtained $M_s$, $\tau_0$, and $\tau_a$ in regulation design.

**Table 3.** Maximum and minimum values of obtained $M_s$.

| $M_s^d$ | Servo | | Regulator | |
| --- | --- | --- | --- | --- |
| | min | max | min | max |
| 1.4 | 1.3923 | 1.4088 | 1.3904 | 1.4216 |
| 1.6 | 1.5836 | 1.6130 | 1.5819 | 1.6183 |
| 1.8 | 1.7725 | 1.8256 | 1.7738 | 1.8266 |
| 2.0 | 1.9518 | 2.0359 | 1.9527 | 2.0356 |

*4.4. Algorithm*

The proposed design procedure is summarized as the following algorithm:

1. The plant parameters and the controller parameters are normalized.
2. For the normalized systems, the constrained optimization problem is solved such that the reference and disturbance responses are optimized, respectively, and the optimal controller parameters are obtained.
3. Based on the obtained optimized parameters, the tuning rule for the controller parameters is designed.
4. Using the tuning rule, the practical PID parameters are decided.

When the tuning rule is obtained once, the PID parameters are calculated without solving the constrained optimization problem.

## 5. Numerical Examples

First, in Section 5.1, the trade-off design between the servo and regulator design is confirmed. The accomplished robust stability is then shown in Section 5.2. Finally, the proposed method is compared with two conventional discrete time trade-off design methods [27,28] in Sections 5.3 and 5.4, respectively.

*5.1. Trade-off Tracking Performance Comparison*

As a controlled plant, the following transfer function is used:

$$P_d(z^{-1}) = \frac{0.0231 + 0.0114z^{-1}}{1 - 0.9753}z^{-14} \tag{36}$$

The transfer function is the discrete time representation of the following continuous time system with $T_s = 0.03$ s:

$$P(s) = \frac{1.4}{1.2s + 1}e^{-0.4s} \tag{37}$$

Using the proposed method, the PID parameters are decided based on Equation (36) with $M_s^d \in \{1.4, 1.6, 1.8, 2.0\}$. The simulation results obtained using the PID parameters are shown in Figure 6, where the reference input is 1.0 and the control input is disturbed by a unit step signal after 15 s. Figure 6 shows that the servo design is superior to the regulator design with respect to reference tracking performance. On the other hand, the regulator design is superior to the servo design with respect to disturbance rejection.

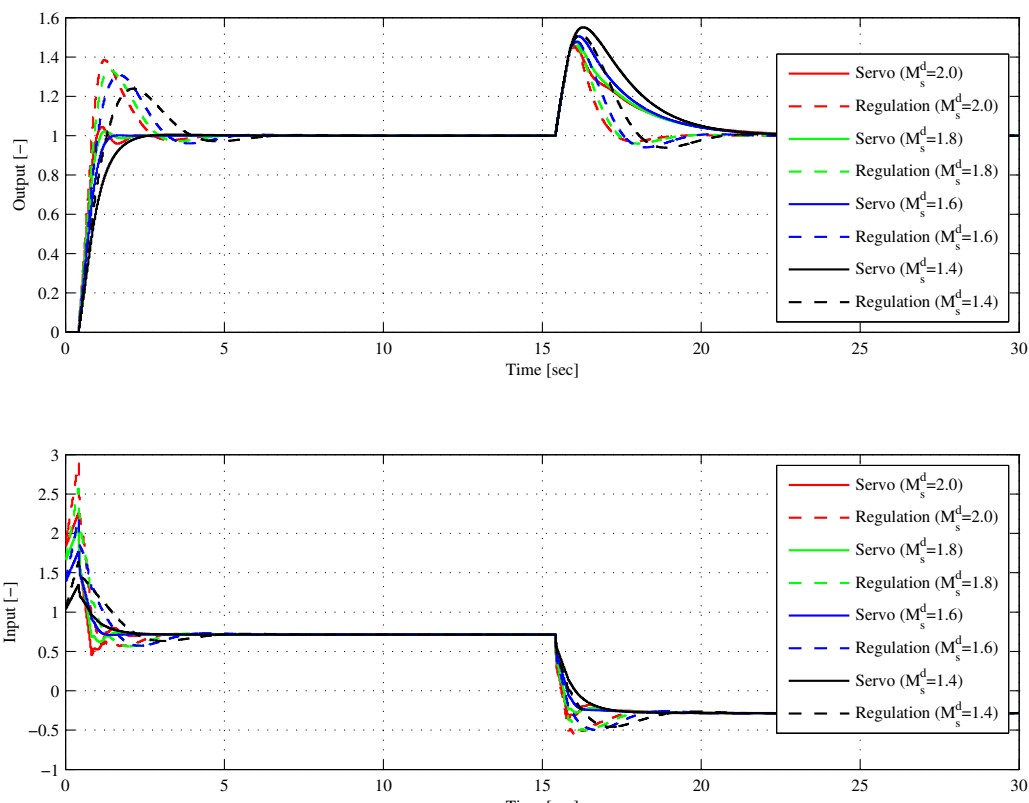

**Figure 6.** Output and input responses obtained using the proposed method in both servo and regulator design, where the system is disturbed by a unit step disturbance after 15 s.

The control performance is evaluated using $J_s$ and $J_r$, where $J_s$ denotes the SAE value from the start until 15 s, and $J_r$ also denotes the SAE value from 15 s until the end. Here, $J_s$ and $J_r$, as well as the PID parameters and $M_s$ in the servo and regulator, respectively, are summarized in Table 4. In the servo design, $J_s$ is smaller than $J_r$. In the regulator design, $J_r$ is smaller than $J_s$.

**Table 4.** Proportional-integral-derivative parameter, $M_s$, and index values.

|  | $M_s^d$ | $K_p$ | $T_i$ | $T_d$ | $M_s$ | $J_s$ | $J_r$ |
|---|---|---|---|---|---|---|---|
| | 1.4 | 1.0217 | 1.3331 | 0.1048 | 1.3998 | 0.9576 | 1.2253 |
| Servo | 1.6 | 1.3709 | 1.4633 | 0.1090 | 1.5964 | 0.7638 | 1.1531 |
| Optimization | 1.8 | 1.6359 | 1.5879 | 0.1360 | 1.7937 | 0.7064 | 1.0688 |
| | 2.0 | 1.8093 | 1.7116 | 0.1537 | 1.9936 | 0.6970 | 1.0274 |
| | 1.4 | 1.0159 | 0.6876 | 0.1737 | 1.4052 | 1.3048 | 0.8667 |
| Regulator | 1.6 | 1.3430 | 0.6641 | 0.1681 | 1.5944 | 1.0673 | 0.6466 |
| Optimization | 1.8 | 1.6065 | 0.7020 | 0.1597 | 1.7913 | 0.9705 | 0.5302 |
| | 2.0 | 1.8217 | 0.7174 | 0.1589 | 1.9922 | 0.9458 | 0.4565 |

From both Figure 6 and Table 4, the larger $M_s^d$, the better the control performance. Therefore, the trade-off design is achieved using the proposed method.

*5.2. Robust Stability*

The effectiveness of the accomplished robust stability is shown. The PID controller is designed based on the nominal model Equation (36) in Section 5.1 in the servo and regulation, respectively.

The scenario is that the plant is Equation (36) from the start to 30 s and is changed to Equation (38) after 30 s as the model perturbation.

$$P'_d(z^{-1}) = \frac{0.0530}{1 - 0.9788}z^{-21} \tag{38}$$

The transfer function is the discrete time version of the following continuous time model with the sampling interval $T_s = 0.03$ s:

$$P'(s) = \frac{2.5}{1.4s + 1}e^{-0.6s} \tag{39}$$

The simulation results for the servo and regulator designs are shown in Figure 7, where the reference input is given by a unit step function, and the control input is disturbed by a unit step function after 15 s. The simulation results show that the effect of the model perturbation is suppressed by small $M_s^d$. However, note that the control performance is superior before the model perturbation when the value of $M_s^d$ is large.

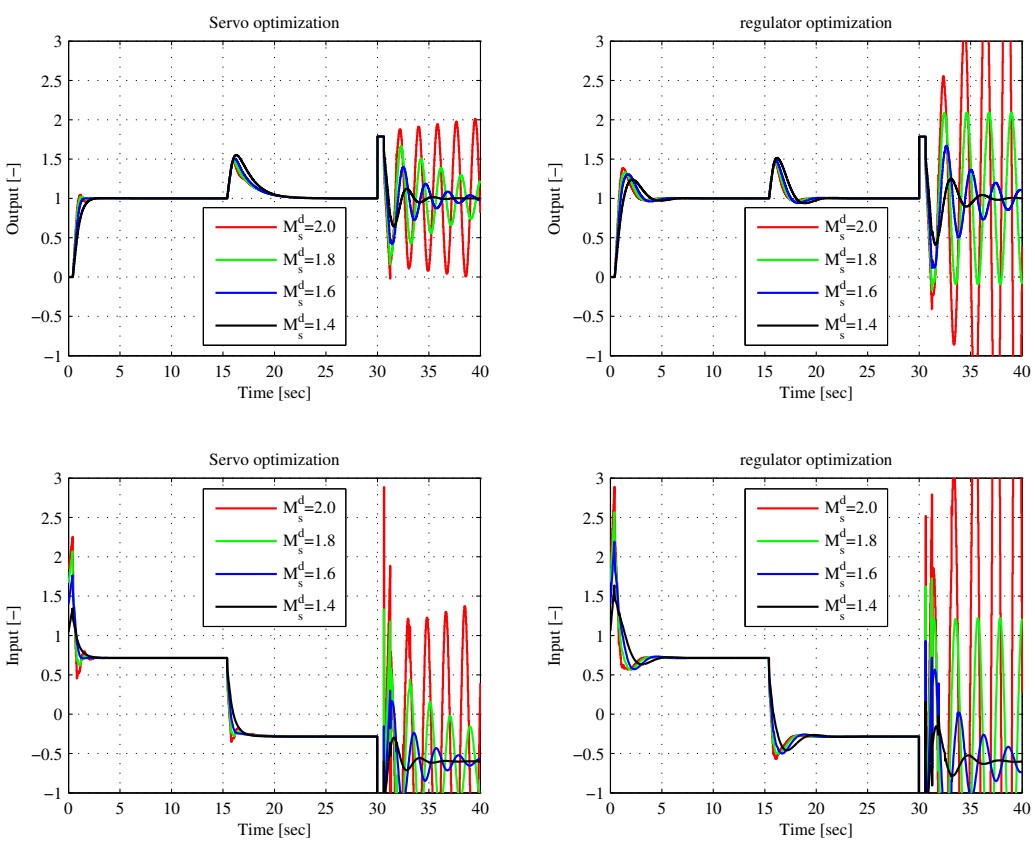

**Figure 7.** Output and input responses obtained using the proposed method for each $M_s^d$ with model perturbation, where the system is disturbed after 15 s; left: servo optimization; right: regulator optimization.

### 5.3. Comparison with the IMC-Based Method

Consider the following discrete time system:

$$P_d(z^{-1}) = \frac{0.0100}{1 - 0.99z^{-1}}z^{-25} \tag{40}$$

The system is the discrete time representation of the following continuous time system with a sampling interval of 0.01 s:

$$P(s) = \frac{1}{s+1} e^{-0.25s} \tag{41}$$

In the conventional discrete time trade-off design method [27], the IMC controller is designed as:

$$C_{IMC}(z^{-1}) = \frac{(1-a_1)(1-\lambda_c)}{b_0(1-\lambda_c z^{-1})} \tag{42}$$

where $\lambda_c$ is the trade-off design parameter. The obtained IMC controller is approximated by the following discrete time PID control law:

$$C_{PID}(z^{-1}) = K_p \left\{ 1 + \frac{1}{T_i(1-z^{-1})} + (1-\alpha T_d)\frac{T_d(1-z^{-1})}{1-\alpha T_d z^{-1}} \right\} \tag{43}$$

where $\lambda_c$ is designed in the range of [0.8,0.99]. However, the designed control systems are unstable when $\lambda_c \leq 0.96$, and thus, the conventional control law is designed such that $\lambda_c$ is set to 0.97, 0.98, and 0.99, respectively, where $\alpha = 0.1$. The designed PID parameters for the conventional method and the proposed method are shown in Table 5.

**Table 5.** Proportional-integral-derivative parameters, $M_s$, and sum of absolute errors (SAE) values using the proposed and internal model control (IMC)-based methods for Equation (40).

|  |  | $K_p$ | $T_i$ | $T_d$ | $M_s$ | $J_s$ | $J_r$ |
|---|---|---|---|---|---|---|---|
| IMC-based method [27] | $\lambda_c = 0.99$ | 0.8248 | 103.10 | 4.6696 | 1.1625 | 1.2500 | 1.2493 |
|  | $\lambda_c = 0.98$ | 1.3978 | 104.83 | 7.5346 | 1.2937 | 0.7533 | 0.7500 |
|  | $\lambda_c = 0.97$ | 1.8184 | 106.07 | 9.4658 | 1.5659 | 0.5919 | 0.5833 |
| Proposed Servo | $M_s^d = 1.4$ | 1.9120 | 1.1242 | 0.0606 | 1.4014 | 0.5936 | 0.5878 |
|  | $M_s^d = 1.6$ | 2.5724 | 1.2107 | 0.0690 | 1.5919 | 0.4733 | 0.4705 |
|  | $M_s^d = 1.8$ | 3.0641 | 1.3062 | 0.0871 | 1.7869 | 0.4367 | 0.4260 |
|  | $M_s^d = 2.0$ | 3.3823 | 1.4277 | 0.0984 | 1.9851 | 0.4313 | 0.4216 |
| Proposed Regulator | $M_s^d = 1.4$ | 1.9193 | 0.5000 | 0.1070 | 1.3999 | 0.8079 | 0.3352 |
|  | $M_s^d = 1.6$ | 2.5319 | 0.4605 | 0.1079 | 1.5858 | 0.7646 | 0.2432 |
|  | $M_s^d = 1.8$ | 3.0234 | 0.4928 | 0.1038 | 1.7778 | 0.6951 | 0.1975 |
|  | $M_s^d = 2.0$ | 3.4407 | 0.5012 | 0.1069 | 1.9940 | 0.6642 | 0.1689 |

The simulations were conducted using the conventional and proposed methods, where the reference input was set to 1.0, and the control input was disturbed by a unit step function after 30 s. The conventional method is compared with the proposed servo and regulator optimization methods in Figure 8. The obtained $M_s$ values and the evaluated values $J_s$ and $J_r$ are also summarized in Table 5, where $J_s$ denotes the SAE value from the start until 30 s, and $J_r$ also denotes the SAE value from 30 s until the end. Table 5 shows that the conventional method provides a trade-off design by selecting $\lambda_c$ even though no value is assigned to $M_s$. Moreover, the tracking performances obtained using the proposed method are superior to those obtained using the conventional method.

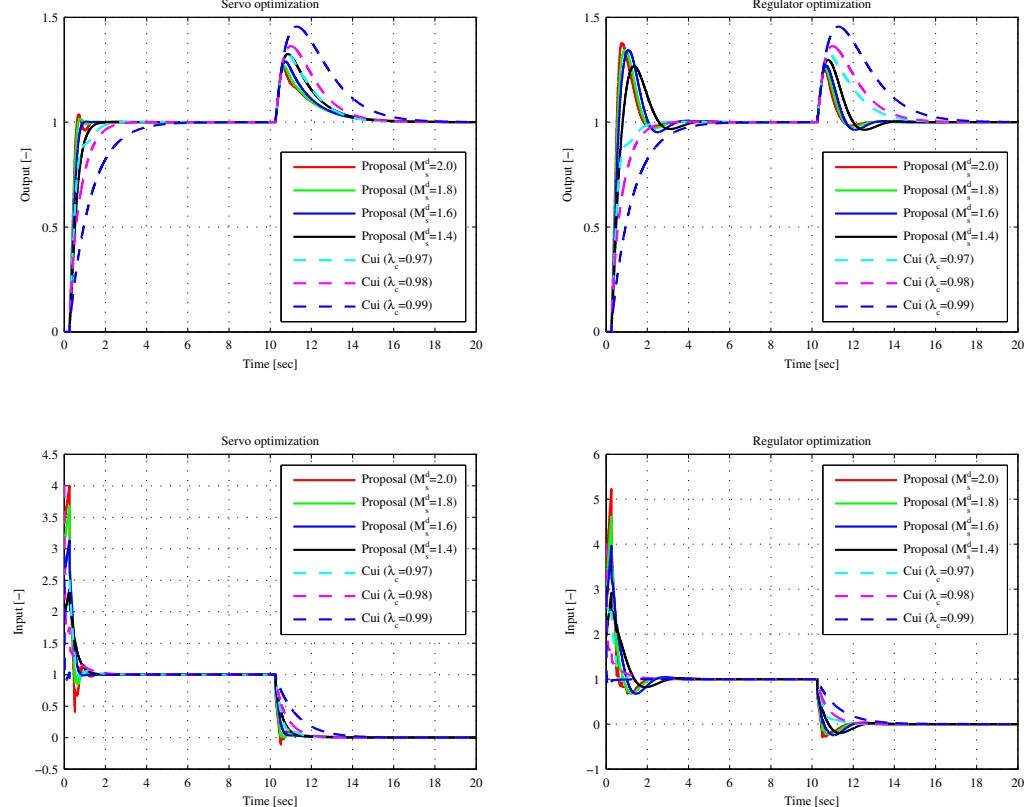

**Figure 8.** Results of the IMC-based method and the proposed servo and regulator optimization methods, where the system is disturbed by a unit step disturbance after 10 s; left: servo optimization; right: regulator optimization.

### 5.4. Comparison with the Conventional Discrete-Time Method

The conventional discrete time design method [28] is compared with the proposed method. Here, two scenarios are conducted, in which a non-zero plant and a zero-included plant, respectively, are controlled.

In the first simulation, we consider the following non-zero discrete time plant:

$$P_{d1}(z^{-1}) = \frac{0.05127z^{-1}}{1 - 0.9486z^{-1}} z^{-10} \tag{44}$$

where Equation (44) is the discrete time representation of Equation (45) with a sampling interval of $T_s = 0.05$ s.

$$P_1(s) = \frac{1}{0.95s + 1} e^{-0.5s} \tag{45}$$

Equation (44) has no zero since the continuous time dead-time is an integer multiple of the sampling interval.

In the second simulation, the controlled discrete time system is given as follows:

$$P_{d2}(z^{-1}) = \frac{0.0201 + 0.02473z^{-1}}{1 - 0.9552z^{-1}} z^{-7} \tag{46}$$

where Equation (46) is the discrete time version of the continuous time system given by Equation (47) with a sampling interval of $T_s = 0.061$ s.

$$P_2(s) = \frac{1}{1.33s + 1} e^{-0.4s} \tag{47}$$

Equation (46) has a zero since the dead-time in the continuous time model is not an integer multiple of the sampling interval. Since the discrete time system has a zero, the conventional method is not directly used. Therefore, Equation (46) is hereby approximated by the next discrete time system, and the conventional method is used:

$$P'_{d2}(z^{-1}) = \frac{0.04483z^{-1}}{1 - 0.9552z^{-1}} z^{-7} \tag{48}$$

In the simulations, the reference input is set to 1.0. Furthermore, the control input is disturbed by a unit step function signal after 10 s. The obtained PID parameters are shown in Tables 6–9. Using the obtained parameters, the discrete time models Equations (44) and (46) are controlled, respectively, and the output results are plotted in Figures 9 and 10. Furthermore, the obtained $M_s$ value and index values $J_s$ and $J_r$ are also shown in Tables 6–9, where $J_s$ denotes the SAE value while the control is not disturbed, and $J_r$ denotes the SAE value while the control input is disturbed.

**Table 6.** Proportional-integral-derivative parameters, $M_s$, and SAE values in the servo design using Equation (44).

|  | $M_s^d$ | $K_p$ | $T_i$ | $T_d$ | $M_s$ | $J_s$ | $J_r$ |
|---|---|---|---|---|---|---|---|
| | 1.4 | 0.9373 | 1.0470 | 0.1445 | 1.4002 | 1.1737 | 1.1171 |
| Proposed | 1.6 | 1.2529 | 1.1866 | 0.1346 | 1.6009 | 0.9489 | 0.9465 |
| method | 1.8 | 1.5008 | 1.2976 | 0.1649 | 1.8023 | 0.8833 | 0.8638 |
| | 2.0 | 1.6595 | 1.3763 | 0.1867 | 2.0043 | 0.8703 | 0.8282 |
| | 1.4 | 0.9732 | 1.0888 | 0.1367 | 1.4224 | 1.1427 | 1.1186 |
| Conventional | 1.6 | 1.2996 | 1.2398 | 0.1376 | 1.6311 | 0.9539 | 0.9531 |
| method | 1.8 | 1.5332 | 1.3585 | 0.1682 | 1.8363 | 0.8955 | 0.8846 |
| | 2.0 | 1.6849 | 1.4374 | 0.1894 | 2.0442 | 0.8802 | 0.8513 |

**Table 7.** Proportional-integral-derivative parameters, $M_s$, and SAE values in the regulator design using Equation (44).

|  | $M_s^d$ | $K_p$ | $T_i$ | $T_d$ | $M_s$ | $J_s$ | $J_r$ |
|---|---|---|---|---|---|---|---|
| | 1.4 | 0.9239 | 0.6663 | 0.2190 | 1.4009 | 1.3680 | 0.8922 |
| Proposed | 1.6 | 1.2254 | 0.6837 | 0.2020 | 1.6017 | 1.2768 | 0.6949 |
| method | 1.8 | 1.4660 | 0.7268 | 0.1886 | 1.8026 | 1.2042 | 0.5834 |
| | 2.0 | 1.6614 | 0.7538 | 0.1796 | 2.0054 | 1.1653 | 0.5092 |
| | 1.4 | 0.9608 | 0.6652 | 0.2151 | 1.4227 | 1.3553 | 0.8618 |
| Conventional | 1.6 | 1.2648 | 0.6865 | 0.2002 | 1.6322 | 1.2685 | 0.6745 |
| method | 1.8 | 1.4886 | 0.7154 | 0.1818 | 1.8314 | 1.2143 | 0.5733 |
| | 2.0 | 1.7045 | 0.7580 | 0.1789 | 2.0559 | 1.1583 | 0.4945 |

**Table 8.** Proportional-integral-derivative parameters, $M_s$, and SAE values in the servo design using Equations (46) and (48), respectively.

|  | $M_s^d$ | $K_p$ | $T_i$ | $T_d$ | $M_s$ | $J_s$ | $J_r$ |
|---|---|---|---|---|---|---|---|
| Proposed method | 1.4 | 1.4664 | 1.4390 | 0.1009 | 1.4026 | 1.0088 | 0.9796 |
|  | 1.6 | 1.9725 | 1.5788 | 0.1066 | 1.6010 | 0.8020 | 0.7978 |
|  | 1.8 | 2.3577 | 1.7146 | 0.1354 | 1.8014 | 0.7412 | 0.7236 |
|  | 2.0 | 2.6043 | 1.8463 | 0.1550 | 2.0010 | 0.7320 | 0.7037 |
| Conventional method [28] | 1.4 | 1.3144 | 1.5050 | 0.1075 | 1.3571 | 1.1449 | 1.1407 |
|  | 1.6 | 1.7619 | 1.6841 | 0.1194 | 1.5024 | 0.9553 | 0.9495 |
|  | 1.8 | 2.0745 | 1.8297 | 0.1473 | 1.6576 | 0.8813 | 0.8737 |
|  | 2.0 | 2.2804 | 1.9491 | 0.1662 | 1.8076 | 0.8537 | 0.8441 |

**Table 9.** Proportional-integral-derivative parameters, $M_s$, and SAE values in the regulator design using Equation (46) and (48), respectively.

|  | $M_s^d$ | $K_p$ | $T_i$ | $T_d$ | $M_s$ | $J_s$ | $J_r$ |
|---|---|---|---|---|---|---|---|
| Proposed method | 1.4 | 1.4332 | 0.7274 | 0.1790 | 1.4026 | 1.2958 | 0.6457 |
|  | 1.6 | 1.8980 | 0.7008 | 0.1744 | 1.6030 | 1.2111 | 0.4786 |
|  | 1.8 | 2.2724 | 0.7352 | 0.1645 | 1.8028 | 1.1244 | 0.3915 |
|  | 2.0 | 2.5759 | 0.7527 | 0.1659 | 2.0076 | 1.0778 | 0.3375 |
| Conventional method [28] | 1.4 | 1.2922 | 0.7535 | 0.1982 | 1.3571 | 1.3330 | 0.7262 |
|  | 1.6 | 1.6990 | 0.7295 | 0.1833 | 1.5124 | 1.2267 | 0.5433 |
|  | 1.8 | 2.0191 | 0.7328 | 0.1731 | 1.6620 | 1.1631 | 0.4495 |
|  | 2.0 | 2.2923 | 0.7810 | 0.1659 | 1.8111 | 1.0886 | 0.3925 |

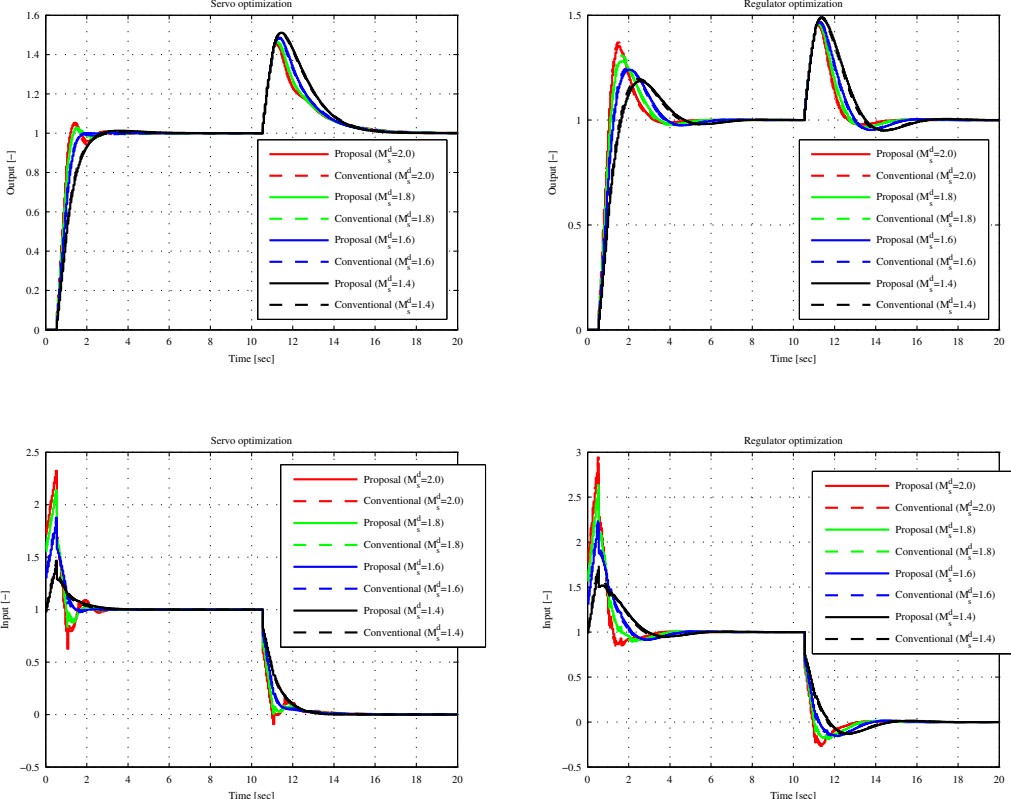

**Figure 9.** Output and input trajectories for the proposed and conventional designs for Equation (44), where the system is disturbed by a unit step disturbance after 10 s; left: servo optimization; right: regulator optimization.

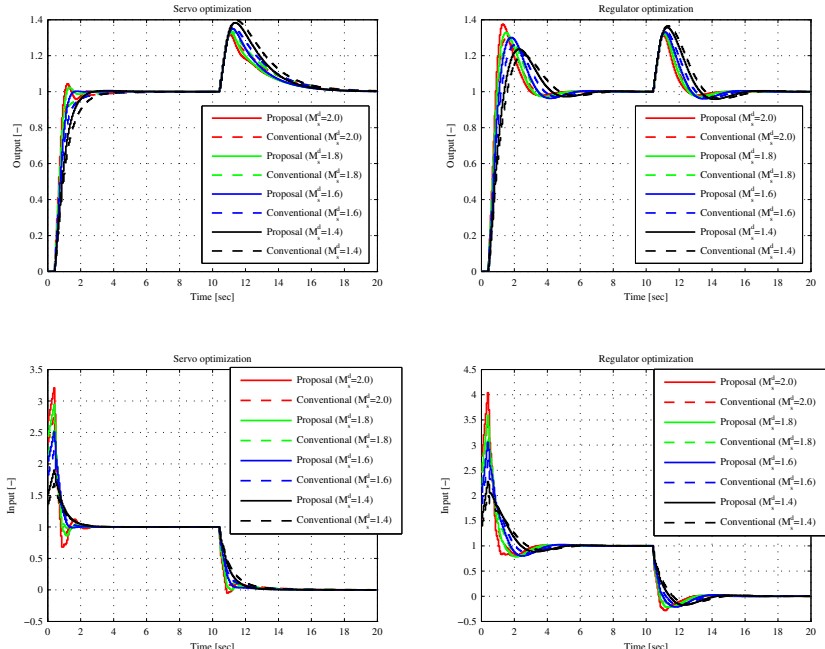

**Figure 10.** Output and input trajectories for the proposed and conventional designs for Equation (46), where the system is disturbed by a unit step disturbance after 10 s; left: servo optimization; right: regulator optimization.

In the case of the non-zero system Equation (44), $J_s$, $J_r$, and $M_s$ obtained using the conventional method were close to those obtained using the proposed method. On the other hand, when the controlled plant was a zero-included system, Equation (46) was out of range of the conventional method. Therefore, the tracking performances using the conventional method were inferior to those using the proposed method, and hence, the SAE values using the conventional method were larger than those using the proposed method. Furthermore, the prescribed robust stabilities $M_s^d$ were achieved using the proposed method, while on the other hand, the $M_s$ values obtained using the conventional method were insufficient.

The simulation results showed that both the conventional and proposed methods were useful for the non-zero plants. However, when the zero-included plant was controlled, the design objective was achieved using the proposed method even though the conventional method was not available. Therefore, the proposed method is a more general method of the conventional discrete time design method.

## 6. Conclusions

The present study proposed a new trade-off PID control design method for discrete time FOPDT systems including a zero. In the proposed method, the regulator- or servo-optimal PID controller was designed in discrete time. In the proposed method, since robust stability was a design parameter, it was adjustable depending on the model perturbation.

In the conventional discrete time design method [28], the designable class was restricted such that the dead-time in the continuous time system must be an integer multiple of the sampling interval, and hence, no zero appeared. On the other hand, in the proposed method, the constraint condition was relaxed.

**Author Contributions:** Conceptualization, R.K. and T.S.; methodology, R.K., T.S., and R.V.; software, R.K.; validation, T.S. and R.V.; formal analysis, R.K.; investigation, T.S. and R.V.; resources, R.K.; data curation, R.K. and T.S.; writing, original draft preparation, T.S.; writing, review and editing, R.V.; visualization, R.K., T.S., and R.V.; supervision, T.S. and R.V.; project administration, T.S., R.V., and Y.K.; funding acquisition, T.S., R.V., and Y.K.

**Acknowledgments:** The present study was supported by JSPS KAKENHI Grant Number 16K06425. This research is supported by the Catalan Government under Project 2017 SGR 1202 and also by the Spanish Government under Project DPI2016-77271-R co-funded with European Union ERDF funds.

**Conflicts of Interest:** The authors declare no conflict of interest.

## Appendix A. Derivation of Normalization Parameters

In the same way as the conventional continuous time design, the normalization parameters are obtained.

Here, $\tau_0$ is derived as follows:

$$
\begin{aligned}
\tau_0 &= \frac{L}{T} \\
&= \frac{T_s}{T} d + \frac{L_0}{T}
\end{aligned}
\tag{A1}
$$

In this equation, $\frac{T_s}{T}$ is derived using Equation (3):

$$
\frac{T_s}{T} = -\log a_1
\tag{A2}
$$

and $\frac{L_0}{T}$ is also derived by eliminating $K$ from Equations (4) and (5):

$$
\frac{L_0}{T} = \log \left( \frac{b_0 + a_1}{(b_0 + b_1)a_1} \right)
\tag{A3}
$$

Equations (A2) and (A3) are substituted into Equation (A1), and Equation (18) is then obtained. From Equation (A2), $\tau_a$ is derived as:

$$
\begin{aligned}
\tau_a &= \frac{T_s}{T} \\
&= -\log a_1
\end{aligned}
\tag{A4}
$$

Since $K$ is the steady-state gain, $\kappa_p$ is obtained as:

$$
\begin{aligned}
\kappa_p &= K K_p \\
&= \frac{b_0 + b_1}{1 - a_1} K_p
\end{aligned}
\tag{A5}
$$

From Equation (A2), $\tau_i$ and $\tau_d$ are calculated as follows:

$$
\begin{aligned}
\tau_i &= \frac{T_i}{T} \\
&= -\frac{T_i \log a_1}{T_s}
\end{aligned}
\tag{A6}
$$

$$
\begin{aligned}
\tau_d &= \frac{T_d}{T} \\
&= -\frac{T_d \log a_1}{T_s}
\end{aligned}
\tag{A7}
$$

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
