# Peer review of "Discrete-Time First-Order Plus Dead-Time Model-Reference Trade-off PID Control Design"

_applsci, doi:10.3390/app9163220_

Round 1

Reviewer 1 Report

I think this paper can be published this journal. 

The following is my comment: Can the proposed approach extend to other types of controller design?

Author Response

I think this paper can be published this journal.

 Authors do thank the reviewer for the positive view of the submitted work

The following is my comment: Can the proposed approach extend to other types of controller design?

It is the author’s opinion that the approach can in fact be extended to other kinds of controller design. The cornerstone of the approach are the cost functionals (16) and (17) in the original manuscript that provide a measure of the performance for the two considered modes of operation (servo and regulation). Here a SAE measure is adopted. However, any other kind of performance metric could be used. Along the same lines, the robustness constraint is explicated here in terms of the maximum sensitive peak (14). However, other robustness measures could also be valid options. Any kind of suitable combination of performance and robustness measures can therefore be used along the same lines of the proposal presented here.

In this work, authors choice for SAE and maximum sensitive peak comes because of the wide popularity and extensive use of these metrics on the process control research community.

Reviewer 2 Report

 This paper proposed a new trade-off PID control design method for discrete-time FOPDT systems including a zero. The PID controller is designed in discrete time and since robust stability is a design parameter, it is adjustable depending on the model perturbation.

From the reviewer point of view, the paper is very well written, the bibliography review is broad, the objectives are clear, the applied methodology and simulation results seem to be interesting.  Furthermore, these simulations help to understand the complexity of the problem and the goodness of the proposal solution.

Even though the quality of the document is high, the reviewer has a question to clarify in order to improve it: The algorithm in section 4.4 does not propose a new tuning formula but a optimization procedure. Then, it is not clear the confusing normalization parameter.  If user must optimize, perhaps the best way is to do it directly without this normalization procedure. Authors must clarify this.

Author Response

This paper proposed a new trade-off PID control design method for discrete-time FOPDT systems including a zero. The PID controller is designed in discrete time and since robust stability is a design parameter, it is adjustable depending on the model perturbation.

From the reviewer point of view, the paper is very well written, the bibliography review is broad, the objectives are clear, the applied methodology and simulation results seem to be interesting.  Furthermore, these simulations help to understand the complexity of the problem and the goodness of the proposal solution.

Authors do thank the reviewer for the positive view of the submitted work

Even though the quality of the document is high, the reviewer has a question to clarify in order to improve it: The algorithm in section 4.4 does not propose a new tuning formula but a optimization procedure. Then, it is not clear the confusing normalization parameter.  If user must optimize, perhaps the best way is to do it directly without this normalization procedure. Authors must clarify this.

Authors understand the reviewer claim. The important points of the normalization are, at least, the following two:

From the process point of view, it allows the optimisation problem to be stated in terms of just two parameters. This simplifies the analysis and elaboration of coefficient parameters. From the controller point of view, the fact of having normalized parameters allows guarantee for obtaining controller tunings that are independent of the process gain and time constant. The expressions that are used here are the corresponding ones in discrete time.
The benefits of using normalized forms is exposed in Reference [9] of the original manuscript.